# Work-Related Quality of Life among Physicians in Poland: A Cross-Sectional Study

**DOI:** 10.3390/healthcare12131344

**Published:** 2024-07-05

**Authors:** Krzysztof Jakimów, Jakub Ciesielka, Monika Bonczek, Joanna Rak, Magdalena Matlakiewicz, Karolina Majewska, Katarzyna Gruszczyńska, Mateusz Winder

**Affiliations:** 1Students’ Scientific Society, Department of Radiology and Nuclear Medicine, Medical University of Silesia, 40-752 Katowice, Poland; jakubciesielka@gmail.com (J.C.); monika.bonczek10@gmail.com (M.B.); joanna.rak01@gmail.com (J.R.); matlakiewiczmagdalena@gmail.com (M.M.); 2Department of Digestive Tract Surgery, Faculty of Medical Sciences in Katowice, Medical University of Silesia, 40-055 Katowice, Poland; majewskakarolina1008@gmail.com; 3Department of Radiology and Nuclear Medicine, Medical University of Silesia, 40-752 Katowice, Poland; kgruszczynska@sum.edu.pl

**Keywords:** quality of life, psychological well-being, psychosocial functioning, professional burnout

## Abstract

Background and objectives: Working in a healthcare setting is associated with high levels of stress and burnout syndrome. Work-related quality of life (WRQoL) remains insufficiently evaluated among physicians. The aim of this study is to assess the WRQoL among physicians of interventional, non-interventional, and diagnostic specialties in Poland. Materials and Methods: Standardized and anonymous WRQoL questionnaires have been filled in by 257 physicians working in Silesia, Poland. After the removal of missing data, 246 individuals were stratified in terms of specialties into the appropriate categories, including interventional, non-interventional, and diagnostics. These categories were compared using the following subscales: general well-being (GWB), home–work interface (HWI), job and career satisfaction (JCS), control at work (CAW), working conditions (WCS), and stress at work (SAW). Results: Out of 246 individuals, 132 were women (53.7%) and 112 (45.5%) were men. There were no differences in terms of WRQoL scores (*p =* 0.220) and subscales GWB (*p =* 0.148), HWI (*p =* 0.368), JCS (*p =* 0.117), CAW (*p =* 0.224), WCS (*p =* 0.609), SAW (*p =* 0.472) between interventional, non-interventional, and diagnostic specialties. The group of young doctors (age ≤ 30 years) had higher JCS scores than the older ones (mean score [SD], 22.7 [3.98] vs. 21 [4.6]; *p* = 0.013). Physicians who were not working in hospital had higher WRQoL score than respondents working in hospital (*p* = 0.061), with significant differences in terms of GWB (mean score [SD], 20.3 [4.93] vs. 22.8 [3.2], *p* = 0.014), HWI (mean score [SD], 9.1 [=2.65] vs. 10.6 [2.73], *p* = 0.011), and WCS (mean score [SD], 9.5 [2.61] vs. 10.8 [2.54], *p* = 0.035). Conclusion: There were no differences considering overall WRQoL between analyzed groups stratified according to specialty. However, we disclosed a significant association between the respondent’s WRQoL and age as well as place of work.

## 1. Introduction

Healthcare workers are exposed to stress and highly prone to burnout syndrome which may affect up to 71% of them depending on the specialty [1]. Stress also has an impact on the educational process, patient care, and quality of life [2,3,4]. Allergies, high blood pressure, heart diseases, and diabetes are among stress-related diseases that may affect over three in five physicians, decreasing their quality of life [5].

Quality of life was defined by the World Health Organization as “individuals’ perceptions of their position in life in the context of the culture and value systems in which they live and in relation to their goals, expectations, standards and concerns” [6]. However, there is no consensus regarding work-related quality of life (WRQoL), which is generally described as the well-being of the employees [4].

The WRQoL Scale is a verified tool that evaluates the quality of life and helps identify both well-organized zones and spaces for improvement where an intervention may be implemented [4,7]. The Scale measures the quality of various areas of life, including working conditions, the impact of the chosen profession on private life, and the level of stress [7].

To assess the WRQoL among healthcare workers, we conducted a survey among doctors working in hospitals in southern Poland. Additionally, the perception of WRQoL among physicians of interventional, non-interventional, and diagnostic specialties was compared.

## 2. Objectives

The aim of this study is to assess the differences in the WRQoL among physicians of interventional, non-interventional, and diagnostic specialties in Silesia, Poland.

## 3. Materials and Methods

### 3.1. Study Design

This cross-sectional study is based on a self-reported questionnaire that was adopted and translated from the WRQoL Scale designed at the Department of Psychology of the University of Portsmouth [7]. The study targeted physicians practicing in Silesia Voivodeship, Poland. Physicians of various specialties from 15 different hospitals were randomly asked to participate in the study. Individuals with less than 1 year of seniority were excluded from the study. After obtaining their informed consent, 257 selected individuals filled in the anonymized questionnaires independently. The questionnaires were collected and systematically evaluated for data completeness. The questionnaires with incomplete WRQoL Scale responses were excluded from the analysis, while the absence of other responses was not an exclusion criterion. After the removal of the records with incomplete WRQoL Scale answers, 246 individuals were finally included in the study. The descriptive data used in the study regarded sex, age, and the detailed type of specialty of the respondent, which was categorized by the authors as interventional, non-interventional, or diagnostic, as well as experience of working in healthcare. General surgery, pediatric surgery, oncological surgery, urology, and anesthesiology were classified as interventional specialties. Non-interventional specialties included internal medicine, pediatrics, cardiology, endocrinology, nephrology, diabetology, gastroenterology, and neurology. Radiology was defined as a diagnostic specialty.

The questionnaire consisted of 24 close-ended questions. The first 23 questions were used to calculate six factors that describe the overall quality of life: general well-being (GWB), home–work interface (HWI), job and career satisfaction (JCS), control at work (CAW), working conditions (WCS), and stress at work (SAW). The negative-to-positive strength of the agreement was measured by the Lickert score and expressed as follows: 1—strongly disagree, 2—disagree, 3—neutral, 4—agree, and 5—strongly agree. Three negatively phrased questions were scored reversely. To distinguish the desired cohorts in terms of GWB, HWI, JCS, CAW, WCS, and SAW, the percentile cut-offs of 30% and 60% were used as suggested by the designers of the scale [7]. 

### 3.2. Statistical Analysis

Statistical analysis was performed using IBM SPSS Statistics 29 software. Qualitative variables were presented as absolute values and percentages. Quantitative variables were presented as ranges and means with standard deviations. The distribution of the variables was tested with the Shapiro–Wilk test. The Chi-square test, Fischer’s exact test, Kruskal–Wallis test, or Mann–Whitney U test were used for between-group comparisons in terms of WRQoL scores. A *p* value of <0.05 was considered significant.

## 4. Results

Out of 257 analyzed cases, 11 respondents were excluded due to incomplete answers for the WRQoL Scale. Among the 246 analyzed participants, there were 132 (53.7%) women and 112 (45.5%) men. In the interventional subgroup, men accounted for 67.8% of the physicians, and 27.7% of the non-interventional subgroup. Most of the respondents (58.1%) were young doctors (below 41 years old), with up to 5 years of working experience (29.3%). However, a similar group of respondents (28.9%) had at least 20 years of experience. The hospital was most frequently identified as the main place of work among all specialties (81.3%). No physician of the diagnostic or interventional specialty worked in general practice. The majority of participants worked in public healthcare (85%). Detailed characteristics of the study population are presented in Table 1.

Most of the respondents represented non-interventional specialties (48.4%), followed by interventional (35.4%) and diagnostic specialties (16.3%). The overall mean (SD) WRQoL score in the analyzed population was 76.1 (15.67). The highest mean score of 77.9 (14.8) was noted in the interventional group, while the lowest was among the diagnostic group (mean score [SD], 73.6 [15.21]). These scores were statistically similar between the analyzed groups (*p =* 0.22). There were also no significant differences between GWB (*p =* 0.15), HWI (*p =* 0.37), JCS (*p =* 0.117), CAW (*p =* 0.22), WCS (*p =* 0.61), and SAW (*p =* 0.47). The comparison of scores derived from the WRQoL Scale between groups with different specialties is presented in Table 2.

Between-group comparisons of WRQoL scores revealed no differences between sex groups (Table 3).

The group of young doctors (age ≤ 30 years) had higher JCS scores than the older ones (mean score [SD], 22.7 [3.98] vs. 21 [4.6]; *p =* 0.013). Respondents with more than 10 years of working experience had higher CAW scores than doctors with less experience (mean score [SD], 9.9 [2.68] vs. 10.6 [2.53]; *p =* 0.009). There was no difference between groups of doctors working in private healthcare and predominantly in public healthcare, however, doctors who were not working in a hospital had higher WRQoL score than respondents working in hospital (*p =* 0.061), with significant differences in terms of GWB (mean score [SD], 20.3 [4.93] vs. 22.8 [3.2], *p =* 0.014), HWI (mean score [SD], 9.1 [=2.65] vs. 10.6 [2.73], *p =* 0.011), and WCS (mean score [SD], 9.5 [2.61] vs. 10.8 [2.54], *p =* 0.035). 

## 5. Discussion

Medical professionals are susceptible to stress and burnout syndrome and can exhibit symptoms of depression from the time they are students [8]. Those conditions may result in impaired patient care, including major medical errors, suboptimal treatment, and decreased patient satisfaction [3,9]. 

The WRQoL Scale consists of six factors: GWB, HWI, JCS, CAW, WCS, and SAW. GWB is a self-describing component that assesses general happiness and satisfaction in life. When GWB is low, it may reduce work performance. Thus, it seems crucial to promote GWB and prevent any circumstances that are able to affect it negatively [7]. Such circumstances include night shifts, high levels of stress, and insufficient sleep [10]. This may be a reason for the lower GWB that was noted in our study in the subgroup of physicians whose main place of work was a hospital compared to others (20.3 vs. 22.8, *p* = 0.014). A study by Zubair et al. [4] showed that some subgroups of interventional specialists were characterized by a better GWB compared to others. They included male physicians, doctors who have children, and physicians with appropriate professional experience [4]. This association was not exhibited in our study, as the interventional group had a similar GWB to other specialties. According to Somsila et al. [5], most residents in northeastern Thailand agreed that their general well-being was average to good. However, the majority of respondents denied or were neutral towards the statement “My life is similar to the ideal life”. Some of the young medical doctors were unhappy or even suffered from depression. We did not disclose any difference in GWB between physicians who were ≤30 years old and those >30 years old (21.6 vs. 20.2, *p* = 0.099). Our study did not reveal an association between GWB and type of specialty or doctor’s gender, indicating a similar level of general happiness and satisfaction in life. 

HWI evaluates the balance between work and an individual’s personal life and how supportive an employer can be to an employee’s home life. Low values of HWI can be due to either the demands of home or the strains of work. The first implies that it is hard for an individual to be at work when necessary, and the result may be inadequate, while the latter means that it is difficult to leave work, which results in diminished satisfaction in other aspects of life [7]. Our study showed that HWI scores were lower in the group that worked mainly in a hospital (9.1 vs. 10.6, *p* = 0.011). This may be caused by more flexible working hours in the private system than in public healthcare; however, this was not checked in our study. Research that was conducted by Putnik et al. [11] showed that work-related tasks had a stronger impact on an individual’s personal life than home roles have on the fulfillment of work duties. HWI depends on having offspring or not and their age because young children demand much more attention and childcare. Doctors with children older than 12 experienced higher HWI than those with younger offspring. Our survey did not include questions about doctors’ offspring or their age, which is one of the limitations of our work and does not allow us to refer to previous observations. In our study, the association between HWI and specialties was not disclosed. 

JCS measures how capable the job is in the case of fulfilling one’s potential or providing a sense of achievement and high self-esteem. This can be generally described as how positive someone feels about their work. These aspects are dependent on one’s individual characteristics and working environment [7]. We noticed the difference between physicians ≤ 30 years old, whose JCS was higher than that of physicians > 30 years of age (22.7 vs. 21, *p* = 0.013). Karaferis et al. [12] assessed the factors that negatively affect JCS. They were described as having insufficient salaries and fringe benefits, limited access to perform operational procedures, poor organizational policies, and a lack of chances to advance in their careers. Respondents positively assessed co-workers, the nature of work, and supervision [12]. Those aspects may play a key role in defining JCS over a longer period, which would explain the differences that we found. Another important aspect is place of employment. Interestingly, hospital and institutional activities can provide physicians with higher job satisfaction than working in the private sector, where the conditions and salaries are usually better. This phenomenon was revealed during a study involving French anesthesiologists whose professional experience was longer than 2 years [13]. However, a study related to the characteristics of the place of work that was conducted by Degen et al. [14] among general practitioners in Germany highlighted that practice owners and employed doctors had higher job satisfaction than physicians working in hospitals. This observation was mainly explained by the level of stress. In our study, despite the differences in the physicians’ place of employment, there were no differences in JCS, indicating comparable levels of satisfaction among physicians in different workplaces. This could be explained by the equally high level of stress and strain on doctors working in the private sector compared to those working in the hospital. Moreover, we did not find any differences in JCS regarding the chosen specialty.

The factor evaluating general control, including the ability to participate in the process of making decisions at work, is defined as CAW. It strongly correlates with employees’ health and well-being [7]. According to Somsila et al. [5], CAW was highest among medical residents in Thailand. Nearly half of them felt that they could make decisions that may affect their workplace or discuss the changes with their superiors [5]. Hashemi et al. [15] stated that CAW has the highest impact on overall quality of life at work. In our study, there were no differences between interventional, non-interventional, and diagnostic specialties considering CAW. This proves that the chosen specialty does not affect the decision-making process regarding changes in the workplace Moreover, place of work (hospital vs. others) and the main system of work have no influence on the ability to make decisions independently. However, physicians with over 10 years of working experience tended to have higher CAW scores than those with shorter experience (9.9 vs. 10.6, *p* = 0.009) which may be explained by the greater autonomy of the physicians who worked longer.

WCS describes dissatisfaction based on meeting an individual’s basic requirements, including working conditions, safety, and level of available resources that are vital to the most effective performance. This factor was derived to assess dissatisfaction at work, contrary to JCS which focuses on improving one’s satisfaction [7]. In our study, the group whose main workplace was a hospital scored lowest in this category (9.5 vs. 10.8, *p* = 0.035). According to Bovier et al. [16], administrative burden contributes to higher work-related dissatisfaction. Physicians who spent less time on administrative work rated their level of satisfaction much higher. On the other hand, we did not disclose any significant differences between the assessed groups depending on the chosen specialty in terms of WCS. This may be explained by the equal access to resources that are necessary for work.

The level of pressure and stress is assessed using SAW. Stress is an adverse reaction to excessive demands and is regulated by an individual’s belief in whether they can cope with those situations [7]. A study conducted on anesthesia teams by van Beuzekom et al. [17] estimated that in this group there were some factors like procedures, material resources, and access to information that were most strongly related to SAW. The same study also revealed that, among anesthesiologists, women reported higher levels of stress than men. According to research by Zubair et al. [4], male and female general surgery residents experience similar amounts of stress at work. However, being a senior resident or being in a relationship diminished stress at work. The results of our study did not reveal significant differences between male and female respondents in terms of SAW (5 vs. 5.2, *p* = 0.368). A comparison between psychiatrists and other medical specialists in Finland was made in the research by Heponiemi et al. [18]. It showed that psychiatrists experienced higher SAW than other medical specialists, particularly patient-related stress, which was assumed to result from aggressiveness among patients. Our study disclosed that scores of SAW were similar between all groups. This finding shows that all specialties are equally affected by stress during working shifts.

In comparison to the results of a survey conducted by Al-Hmaid et al. [19] in 2024 among nurses in Jordan, our study suggests that the WRQoL among physicians is better than among nurses who deal with similar working environments. In both studies, the same questionnaire was implemented. The overall physician’s WRQoL score reached 76.1, whereas nurses scored 70.54. In almost every single category, physicians assessed their situations better than nurses (GWB: 20.5 vs. 18.6; HWI: 9.3 vs. 8.53; JCS: 21.4 vs. 19.61; CAW 10.3 vs. 9.57; and WCS 9.7 vs. 8.64, respectively). Interestingly, the stress at work (SAT) parameter was rated higher by nurses compared to physicians (5.59 vs. 5.1). Considering the following data, physicians’ working conditions are relatively better in relation to other healthcare workers, in this case nurses. However, a potential limitation may be inherent to this analysis. Al-Hmaid et al. [19] included only women in the analysis, whereas our study involved both sexes. Thus, it cannot be ruled out that sex constituted a confounding factor. Additional bias could result from cultural differences.

Another cross-sectional study conducted among medical doctors in Poland in 2022 using the same questionnaire showed that the WRQoL was low [20]. Approximately 40% of respondents assessed their quality of work-life as “bad”, whereas almost 10% defined it as “catastrophic”. Surprisingly, the authors discovered that JCS was relatively low in their study group. One-third of doctors declared feeling of dissatisfaction with their work and the same fraction could not state whether they were satisfied with their work. Respondents listed working conditions, remuneration and additional benefits, implementation of professional training, attitudes of their seniors, and the culture of the organization in their department as factors leading to low JCS [20]. Our study revealed similar levels of JCS amid different types of specialties; however, we noted higher JCS in the group of physicians ≤ 30 years old than among physicians > 30 years old (22.7 vs. 21, *p* = 0.013). In our study participants included both physicians during specialization training and medical specialists, whereas research conducted by Storman et al. [20] included only Polish medical residents. This caused a huge difference in the age range of respondents (from ≤30 up to ≥71 vs. 26–35) which can impact the comparative results. 

In addition, Zgliczyńska et al. [21] in their systematic review associated low WCS factor with a greater risk of burnout syndrome among Polish physicians. The excessive number of working hours, numerous shifts, and a high workload were assumed to be related to the risk of burnout syndrome. Moreover, doctors of particular specialties appeared to be more susceptible to burnout syndrome and its earlier onset both in Poland and worldwide—the following groups of specialists retrieved from the study were anesthesiologists and neurologists. Our results did not confirm any differences in overall WRQoL score and WCS among analyzed medical specialties. It turned out that place of work has a significant impact on the WCS score. Working outside a hospital increases WCS in comparison to physicians working in a hospital (10.8 vs. 9.5, *p* = 0.035). However, a huge disproportion in the size of both groups might be a potential limitation. 

Research conducted in university hospitals in Poland by Dubas-Jakóbczyk et al. [22] measured the satisfaction level among Polish physicians. The study showed that physicians assessed their level of JCS as 4.0 (on a scale from 1.0 to 6.0). Some of the factors included in the study, like salary levels, work-personal life balance, and ability to maintain satisfying non-work-related activities, significantly contributed to lowering the level of JCS. The highest levels of satisfaction were connected with diversity and good clinical conditions of patients along with meeting their needs [22]. Interestingly, our study revealed that in spite of different places of employment, there were no differences in JCS. Our results showed that the age of respondents had an impact on the level of JCS. Among the surveyed doctors, physicians ≤30 years old scored remarkably higher in terms of JCS than physicians >30 years of age (22.7 vs. 21, *p* = 0.013).

According to Walkiewicz et al. [23], Polish medical residents who graduated from the Medical University in Gdańsk presented a higher level of GWB than general society. Respondents were more satisfied with some areas of their lives, such as material aspects, education, life achievements, future prospects, and health. However, there were some aspects of life that negatively affected their quality of life assessment. These included high work-related stress, lack of social support, and lack of contact with close friends. 

In our study, we did not make a comparison between physicians and the general public. However, our results revealed that the type of specialization chosen by the young doctor probably does not affect their GWB. This factor might be decreased among respondents working in hospitals compared to the group working outside hospitals (20.3 vs. 22.8, *p* = 0.014).

In a quantitative cross-sectional survey, Domagała et al. [24] identified the factors associated with higher levels of career satisfaction among physicians working in Polish hospitals. Sex, type of hospital, number of working hours per week, number of years of work experience, as well as the stage of professional development (completed specialty training), were the factors with the strongest impact on the respondents’ WRQoL. These areas may explain our results considering JCS among examined physicians.

To the best of our knowledge, this is the first cross-sectional study that aimed to assess the possible differences between interventional, non-interventional, and diagnostic specialties in Poland using a standardized WRQoL Scale. Considering this, we strongly believe that periodical evaluation of WRQoL among physicians can provide a better insight into working conditions and may result in implementing solutions that would enhance both the quality of life and the effectiveness of the personnel. Improving WRQoL may decrease the prevalence of burnout syndrome and lead to better patient care. However, this study is limited by not taking into consideration factors such as marital status, children, income, night shifts, administrative duties, health status, and free time. The small and mostly unified cohort is also a limiting point.

## 6. Conclusions

There were no significant differences between interventional, non-interventional, and diagnostic specialties when either general WRQoL or specific subscales were considered.

Young doctors (<30 years old) scored higher regarding JCS. Physicians with more than 10 years of working experience showed higher CAW scores. There was no difference between the groups of doctors working in private and public healthcare. However, employment in the private sector influenced higher WRQoL, with significant differences in terms of GWB.

More studies among larger cohorts of medical professionals should be performed in the future to verify this thesis and to assess the changes in WRQoL among physicians.

## Figures and Tables

**Table 1 healthcare-12-01344-t001:** General characteristics of the study population.

Variable		Total Population (*n* = 246)	Diagnostic (*n* = 40, 16.3%)	Interventional (*n* = 87, 35.4%)	Non-Interventional (*n* = 119, 48.4%)	*p*
Age (years)	≤30	55 (22.4%)	7 (17.5%)	19 (21.8%)	29 (24.4%)	0.47
	31–40	88 (35.8%)	14 (35%)	29 (33.3%)	45 (37.8%)	
	41–50	33 (13.4%)	4 (10%)	11 (12.6%)	18 (15.1%)	
	51–60	48 (19.5%)	9 (22.5%)	20 (23%)	19 (16%)	
	61–70	18 (7.3%)	2 (5%)	8 (9.2%)	8 (6.7%)	
	≥71	2 (0.8%)	2 (5%)	0	0	
	Lack of data	2 (0.8%)	2 (5%)	0	0	
Sex	Women	132 (53.7%)	19 (47.5%)	28 (32.2%)	85 (71.4%)	<0.001
	Men	112 (45.5%)	20 (50%)	59 (67.8%)	33 (27.7%)	
	Lack of data	2 (0.8%)	1 (2.5%)	0	1 (0.8%)	
Main place of work	General practitioner	17 (6.9%)	0	0	17 (14.3%)	<0.001
	Specialist clinic	6 (2.4%)	1 (2.5%)	4 (4.6%)	1 (0.8%)	
	Hospital	200 (81.3%)	37 (92.5%)	70 (80.5%)	93 (78.2%)	
	General practitioner + Hospital	3 (1.2%)	0	1 (1.1%)	2 (1.7%)	
	Specialist clinic + Hospital	19 (7.7%)	2 (5%)	12 (13.8%)	5 (4.2%)	
	All	1 (0.4%)	0	0	1 (0.8%)	
Main system of work inhealthcare	Private healthcare	8 (3.3%)	3 (7.5%)	5 (5.7%)	0	0.005
	Public healthcare	209 (85%)	32 (8%)	68 (78.2%)	109 (91.6%)	
	Both	27 (11%)	4 (10%)	14 (16.1%)	9 (7.6%)	
	Lack of data	2 (0.8%)	1 (2.5%)	0	1 (0.8%)	
Years of experience	≤5	72 (29.3%)	11 (27.5%)	26 (29.9%)	35 (29.4%)	0.50
	6–10	41 (16.7%)	9 (22.5%)	9 (10.3%)	23 (19.3%)	
	11–20	61 (24.8%)	8 (20%)	22 (25.3%)	31 (26.1%)	
	≥20	71 (28.9%)	11 (27.5%)	30 (34.5%)	30 (25.2%)	
	Lack of data	1 (0.4%)	1 (2.5%)	0	0	

**Table 2 healthcare-12-01344-t002:** Comparison of quality of life between groups with different specialties.

Variable	Total	Diagnostic (*n* = 40, 16.3%)	Interventional (*n* = 87, 35.4%)	Non-Interventional (*n* = 119, 48.4%)	*p **
Overall WRQoL score	76.1 (29–112, 15.67)	73.6 (29–106, 15.21)	77.9 (32–108, 14.8)	75.6 (33–112, 16.4)	0.22
GWB	20.5 (6–30, 4.84)	19.5 (8–30, 5.09)	21.2 (6–29, 4.67)	20.3 (6–30, 4.85)	0.15
HWI	9.3 (3–15, 2.69)	8.8 (3–15, 2.59)	9.5 (3–15, 2.57)	9.3 (3–15, 2.81)	0.37
JCS	21.4 (7–30, 2.65)	20.6 (7–30, 4.09)	22.1 (11–30, 4.57)	21.1 (7–30, 4.61)	0.12
CAW	10.3 (3–15, 2.65)	9.9 (4–15, 2.29)	10.5 (4–15, 2.67)	10.2 (3–15, 2.75)	0.22
WCS	9.7 (3–15, 2.63)	9.7 (3–15, 2.4)	9.5 (4–15, 2.52)	9.8 (3–15, 2.78)	0.61
SAT	5.1 (2–9, 1.78)	5.2 (2–8, 1.83)	5.2 (2–9, 1.64)	4.9 (2–9, 1.85)	0.47

* *p*—Kruskal–Wallis test. Post hoc analyses were not included as the differences between the analyzed groups were not significant. CAW, control at work; GWB, general well-being; HWI, home–work interference; JCS, job and career satisfaction; SAT, stress at work; WCS, working conditions; WRQoL, Work-Related Quality of Life.

**Table 3 healthcare-12-01344-t003:** Comparison calculated WRQoL scores (SD) between groups classified according to variables other than specialty.

Variable	Sex (Women vs. Men)	Age (≤30 vs. >30)	Years of Experience (≤10 vs. >10)	Main System of Work (Public Healthcare vs. Private Healthcare)	Place of Work (Hospital vs. Others)
Overall WRQoL score	75.5 (15.83) vs. 77.2 (15.19), *p =* 0.404	79.9 (14.43) vs. 75.2 (15.7), *p =* 0.103	75.2 (16.69) vs. 77.2 (14.4), *p =* 0.424	75.6 (16.11) vs. 79.7 (12.05), *p =* 0.167	75.5 (15.71) vs. 82 (14.22), *p =* 0.061
GWB	20.2 (4.75) vs. 20.9 (4.85), *p =* 0.238	21.6 (4.58) vs. 20.2 (4.83), *p =* 0.099	20.5 (5.24) vs. 20.6 (4.4), *p =* 0.975	20.3 (4.96) vs. 21.8 (4.05), *p =* 0.078	20.3 (4.93) vs. 22.8 (3.2), *p =* 0.014
HWI	9.3 (2.81) vs. 9.2 (2.51), *p =* 0.683	9.5 (2.76) vs. 9.2 (2.66), *p =* 0.458	9 (2.87) vs. 9.6 (2.47), *p =* 0.120	9.2 (2.72) vs. 9.7 (2.53), *p =* 0.296	9.1 (2.65) vs. 10.6 (2.73), *p =* 0.011
JCS	21.1 (4.4) vs. 21.8 (4.64), *p =* 0.383	22.7 (3.98) vs. 21 (4.6), *p =* 0.013	21.3 (4.7) vs. 21.5 (4.35), *p =* 0.913	21.3 (4.61) vs. 21.9 (3.82), *p =* 0.539	21.3 (4.55) vs. 22 (4.37), *p =* 0.294
CAW	10.1 (2.69) vs. 10.5 (2.54), *p =* 0.288	10.4 (2.54) vs. 10.3 (2.66), *p =* 0.859	9.9 (2.68) vs. 10.6 (2.53), *p =* 0.009	10.2 (2.66) vs. 10.9 (2.27), *p =* 0.102	10.2 (2.7) vs. 10.9 (2.1), *p =* 0.248
WCS	9.7 (2.62) vs. 9.6 (2.63), *p =* 0.582	10.3 (2.44) vs. 9.5 (2.65), *p =* 0.087	9.6 (2.73) vs. 9.8 (2.52), *p =* 0.625	9.7 (2.66) vs. 9.9 (2.37), *p =* 0.564	9.5 (2.61) vs. 10.8 (2.54), *p =* 0.035
SAT	5 (1.85) vs. 5.2 (1.68), *p =* 0.368	5.4 (1.91) vs. 5 (1.72), *p =* 0.121	5 (1.91) vs. 5.1 (1.65), *p =* 0.656	5 (1.8) vs. 5.5 (1.62), *p =* 0.125	5.1 (1.79) vs. 4.9 (1.68), *p =* 0.640

CAW, control at work; GWB, general well-being; HWI, home–work interference; JCS, job and career satisfaction; SAT, stress at work; WCS, working conditions; WRQoL, Work-Related Quality of Life.

## Data Availability

The original contributions presented in the study are included in the article, further inquiries can be directed to the corresponding authors.

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
