# Peer review of "Work-Related Quality of Life among Physicians in Poland: A Cross-Sectional Study"

_healthcare, 2024, doi:10.3390/healthcare12131344_

Round 1

Reviewer 1 Report

Comments and Suggestions for Authors

I would like to appreciate the work the authors made by writing this document, although its content needs some improvements.

1.  The materials an methods section describes a 247 selected individuals, after removing missing data (line 69), while the results section mentions 246+ participants after removing missing data. What kind of missing data were excluded from the study? Please describe including and exclusion from study criteria in Materials and Methods, as in Table 1 - General characteristics of the study population these is still some missing data (lack of data).

2. The study population, the selected doctors to reply to the questionnaire, is not the ideal case (unequal study groups, most responders work in hospitals - over 90%, private healthcare 85%). The general characteristics of study population could have been more detailed (there are a lot of factors that influence the quality o life among physicians - marital status, children, income, night shifts, free time, etc). Please add limitations of the study.

3. Please verify and correct data in entire Table 1! responders <30years = 7+9+29=45, and in total population in noted 55. Also, please mention what p stands for, what are the groups compared. There are 3 groups to be compared in between, therefor there should be 2 values.

4. Table 2 has also only one p value. What does it compare? as there are 3 groups included in Table 2. Please add details below the table.

5. Table 3. Please add explanations below table or in title about what number are to be found in table 3 (I suppose it is the score obtained in questionnaire). Would it be possible to transform Table 3 in several figures?

6. Discussion section is very long and very vague, nothing concrete.  I understand that you found no statistical significant difference between the groups, but be more specific to  your findings in comparison with other studies. In this point Discussion looks like literature studies review. You can add more data about your own study findings, population and so on. My belief is that you found no differences because of particular study population selection (private healthcare system) and lack of specific factors that really influence the quality of life.

7. In results section, more text should be added in order to understand the main findings found in tables.

8. Only 10 references are newer than 5 years. 

Author Response

Dear Sir/Madam,

Thank You very much for your remarks. We have added suggested improvements to the manuscript. Here are the answers to your questions.

AD. 1. Regarding the number of the records, “246” is the correct number. “247” was an error that was not noted earlier. We are grateful for spotting this.

The questionnaires with incomplete WRQoL Scale responses were excluded from the analysis, while the absence of other responses (age, sex, main system of work and years of experience) was not an exclusion criterion. After the removal of the records with incomplete WRQoL Scale answers, 246 individuals were finally included in the study.

AD. 2. We have stated the limitations at the end of the discussion. We believe this remark is indeed crucial for this paper to be valuable to other researchers.

AD. 3. In Table 1 there was a „1” missing, as 21.8% out of 87 participants in the interventional group is 19 – we are really sorry for this mistake – it has been corrected and now there is <30 years = 7 + 19 + 29 = 55. The p value in Table 1 was calculated in order to describe the general characteristics of the study population and illustrate the differences in the distribution of age, sex, main place of work, main system of work in healthcare and years of experience between the analyzed groups. P value <0.05 means that there is a difference in the distribution of particular variables between the analyzed groups, however post - hoc analyses were not performed, as the analysis of these differences was not the aim of the study.

AD. 4. The P values in the Table 2 were derived from the Kruskal-Wallis test, which compared the three analyzed groups, as the variables had non-normal distribution. Post hoc analyses were not included as the differences between the analyzed groups were not significant. There were no differences between any of the analyzed groups, thus, further analysis was not significant.

AD. 5. We added explanations in the title concerning what number are to be found in table 3. We hope this provides a better insight to the data in the table.

AD. 6. We elaborated more on our data in the Discussion section. We had similar belief regarding the sample, however, our research was based solely on WRQoL Scale and we did not want to add more factors. This paper was meant to be a base for future studies which can implement more disturbing factors. We also feared that long questionnaires would result in even smaller populations, thus, we decided to rely only on the proposed scale.

AD. 7. We added more findings to the text in the Result section. However, it is difficult to directly compare our work with other papers due to different methodologies used. We hope that this change is sufficient.

AD. 8. The main struggle was to find appropriate references. Very few were relevant. This is the main reason that only 10 references are newer than 5 years. We hope that our work would bring more attention to this topic and more articles will be written to cover it.

We hope that our changes will meet your expectations. We truly believe that these remarks were indeed important and our paper can benefit from them.

Kind regards,

The Authors

Reviewer 2 Report

Comments and Suggestions for Authors

The work presented is of great interest and relevance. It is worth paying attention to the quality of life experienced by medical professionals.

In this regard, it is noteworthy that the introduction provides adequate context on stress and burnout in healthcare workers, highlighting the need to assess work-related quality of life (WRQoL) among physicians. The literature review is relevant and cites relevant and current references, although it could benefit from further elaboration on similar previous studies to strengthen the theoretical framework.

On the other hand, the methods are sufficiently detailed, but it would be advisable for the authors to provide a more complete description of the exclusion criteria.

With respect to the results, although they are clearly presented with well-constructed and detailed tables, it would be relevant for the authors to elaborate on the lack of significant differences.

Author Response

Dear Sir/Madam,

Thank You very much for your remarks. We have added suggested improvements to the manuscript. Here are the answers for your questions.

The main struggle was to find the appropriate references. Very few were relevant to our research. We added more findings to the text in the Result section. However, it is difficult to directly compare our work with other papers due to different methodologies used. We hope that this change is sufficient. We hope that our work will bring more attention to this topic and more articles will be written to cover it.

The questionnaires with incomplete WRQoL Scale responses were excluded from the analysis, while the absence of other responses (age, sex, main system of work and years of experience) was not an exclusion criterion. After the removal of the records with incomplete WRQoL Scale answers, 246 individuals were finally included in the study. We added this information to the manuscript.

We elaborated more on our data in the Discussion section. Due to limited papers that assess the quality of life especially among Polish physicians, it is hard to tell why no differences were found. We also stated some limitations that might have had an impact on lack of differences.

We hope that our changes will meet your expectations. We truly believe that those remarks were indeed important and our paper can benefit from them.

Kind regards,

The Authors

Reviewer 3 Report

Comments and Suggestions for Authors

Medical professions are among the so-called "sudden death" professions (Johns JOB 2010, 31(2-3), 361-369). Therefore, the issue of the quality of life of doctors is very important theoretically and practically.

Researchers addressed the issue of doctors' work-related quality of life (WRQoL). Research was conducted on a sample of 246 doctors working in Silesia, one of the most industrial centers of Europe.

The methodological assumptions are correct. The study used a scale proposed by the Department of Psychology at the University of Portsmouth. The examined WRQoL category was composed of the following 6 factors: general well-being (GWB), home-work interface (HWI), job and career satisfaction (JCS), control at work (CAW), working conditions (WCS), and stress at work (SAW). In order to obtain structural results, the following categories of professional specialization of doctors were distinguished: interventional, non-interventional and diagnostic,

The value of the presented research results results from the following facts:

1) The first cross-sectional study in Poland was conducted, the aim of which was to assess possible differences between interventional, non-interventional and diagnostic specializations using the standardized WRQoL Scale. The discussion shows that few studies have been carried out so far in other countries;

2) The results show differences in the development of individual factors, especially between doctors working in hospitals and doctors working in other types of health care units;

3) The division into private and public treatment is not important

4) The type of professional specialty performed by the doctor is not important

5) However, different configurations of factors in particular groups of doctors become more important

The research results are cognitively important. Due to the indicated theoretical values ​​and the practical need to find solutions that minimize stress and burnout, research identifying factors shaping WRQoL should also be extended across professional specialties.

I recommend publishing the article.

Author Response

Dear Sir/Madam,

Thank you very much for your remarks. We have added some minor corrections to the manuscript regarding other Reviewers’ opinions. We believe that work-related quality of life among physicians, especially in Poland, is not assessed properly. We hope that this paper will be the basis for the future studies that will focus on more factors that can improve the quality of life.

We are very grateful for your opinion and hope that our paper met your expectations.

Kind regards,

The Authors

Round 2

Reviewer 1 Report

Comments and Suggestions for Authors

I find the responses of the authors and the changes within the document sufficient for the comments. They added reliable data and pertinent  ideas. I think this document is suitable for publication.